# The Ecological-Dynamics Framework for Medical Skills

**DOI:** 10.3390/healthcare11010038

**Published:** 2022-12-22

**Authors:** Kersi Taraporewalla, André van Zundert, Marcus O. Watson, Ian Renshaw

**Affiliations:** 1The University of Queensland, Brisbane, QLD 4072, Australia; 2Department of Anaesthesia and Perioperative Medicine, Royal Brisbane and Women’s Hospital, The University of Queensland, Brisbane, QLD 4029, Australia; 3School of Psychology, The University of Queensland, Brisbane, QLD 4072, Australia; 4School of Exercise & Nutrition Sciences, Queensland University of Technology, Kelvin Grove, QLD 4000, Australia

**Keywords:** ecological-dynamics, information-movement coupling, active perception, constraints, affordances, movements, medical skills

## Abstract

Physicians are required to move and manipulate equipment to achieve motor tasks such as surgical operations, endotracheal intubations, and intravenous cannulation. Understanding how movements are generated allows for the analysis of performance, skill development, and methods of teaching. Ecological-Dynamics (ECD) is a theoretical framework successfully utilized in sports to explain goal-directed actions and guide coaching and performance analysis via a Constraint-Led Approach (CLA). Its principles have been adopted by other domains including learning music and mathematics. Healthcare is yet to utilize ECD for analyzing and teaching practical skills. This article presents ECD theory and considers it as the approach to understand skilled behavior and developing training in medical skills.

## 1. Introduction

Medical practitioners perform many tasks in caring for patients. Many medical tasks are purely cognitive, as in choosing a particular drug, whilst others are considered psychomotor requiring motor action, such as bronchoscopy or inserting an epidural catheter. Practical tasks in medicine are usually composed of a set of sequential steps each accomplished by movements performed by the practitioner, often assessed by checklists [1]. The fundamental premise regarding all motor tasks is the requirement of voluntary movements, and many involve manipulating equipment to achieve a goal-directed task, such as an endotracheal intubation. Movements require energy transfer generated from forces either external or internal to the agent. Understanding how movements are generated informs the concepts of teaching, practice, assessment, and expertise. Over the last four decades, movement psychologists have adopted concepts of Ecological-Dynamics (ECD) to explain goal-directed actions to overcome the deficiencies of previous cognitive and computational models [2]. This article proposes an ecological-dynamics model of movements as the framework to explain task performance in healthcare, rather than rely on cognitive or computational ones.

## 2. Hypothesis

The hypothesis is ECD be adopted as the framework for understanding movements of practical tasks in medicine.

## 3. The Ecological-Dynamic Explanation of Movements

ECD is a theoretical approach derived from a blend of ecological psychology and dynamical systems theory [3]. It derives ideas from the sub-disciplines of complexity, nonlinear thermodynamics, and synergetics [2]. Originally applied in sports, it has since been applied to multiple domains such as rehabilitation medicine [4,5], physical education [6], golf, music [7], and mathematics [3]. A brief explanation of ecological psychology, dynamical systems theory and complex systems is presented below, prior to exploring ECD.

### 3.1. Ecological Psychology

Ecological psychology (EP) is the study of information transactions between living systems and their environments. Fundamental to EP is the interaction between the agent and the environment, which guides actions and their control [8]. This interaction is dynamic, continuous, and nonlinear as a perceiving-acting cycle forming a feedback loop [8,9,10].

The central tenets of this approach are: (a) direct perception resulting from higher-order invariants; (b) perception as an active process; and (c) biological organisms are specifically sensitive to action possibilities (affordances) afforded by their surroundings [9]. Perception is a continuous process of picking-up information and discriminating different types of situations, resulting from the patterns of energy fluxes created by elements of the world [9,11]. To distinguish between possibilities of information, some lawful structure of the world or environment provides specific invariant energy patterns. For example, during intravenous cannulation, reflected light from the back of the hand provides specific information on the size and location of the vein. Biological organisms do not detect physically defined variables such as the velocity, length, or force applied, but are sensitive to higher-order invariants that give meaning to the sensory input [8]. Higher-order invariants are patterns in ambient energy that specify environmental properties [9]. In catching a ball, attunement to the changing size of the image of the ball on the retina is interpreted as a time-to-contact, the higher order invariant (tau). A driver approaching another car stopped in front of him recognizes the time-to-collision, rather than speed, to apply appropriate force on the brake. In external chest compressions the rescuer is unable to state the magnitude of the force applied during compression but understands it as the resistance felt by the body to decide when to commence recoil.

Perception is an epistemological active process of the knowing of the environment, as the higher-order invariants are distributed across time and space [11,12]. Sensitivity to them is increased by movement. The automobile driver in the example above is actively seeking information regarding speed and distance to the car in front and this is best perceived as his car slows down. The anesthetist inserting an epidural is continuously perceiving the force applied to the plunger of the syringe to detect loss-of-resistance. During bronchoscopy, the proceduralist actively seeks patterns of bifurcations or lack of reflected light (black hole) during movement to recognize the direction to push the scope forward.

Affordances are opportunities or invitations to act provided by the environment to the biological organism in relation to the abilities (effectivities) and intentions of the individual [8,10,12,13,14]. A staircase provides the affordance to an individual intending to move to the floor above. However, a person in a wheelchair does not consider it as an affordance and would be actively seek a ramp as an affordance.

Therefore, human behavior cannot be understood without reference to the environment, and this is on the basis of coupling information and movement [15]. Gibson stressed that this process is continually evolving as a cyclical relationship between perception and movement [16]. The agent perceives what to do by direct perception achieved through specific parameters. This perception of what is possible, is relative to individual abilities (effectivities) and is afforded by the environment.

### 3.2. Dynamical Systems Theory

Dynamics is the study of change in a system over time [12]. The morphology of human behavior and movement can be formalized in terms of low-dimensional variables that vary over time. Such variables include angle between the femur and tibia during locomotion. A set of variables can be generated for a system to describe its state over time. These variables are nonlinear in that equations describing their values are of higher orders. Nonlinear systems display behaviors where variables may approach a fixed value over time irrespective of its starting position. This value is called an attractor [12]. A pendulum without friction approaches a stable period of oscillation depending on its length. The dynamics of complex systems explains many phenomena observed.

### 3.3. Complex Systems

While there is no agreed universal definition of a complex system, a working definition is that it is a special type of system consisting of many parts displaying properties that arise from relations between its components; for example, a group of unorganized elements, such as a pile of stones, is a set. The set can exist without pattern or order, and it behaves as the sum of the behaviors of each element. If the parts are related to each other by an order and organized in a special way, behavior emerges due to local interactions and can be considered a complex system. There are many complex systems in the world. Examples include weather patterns, a car engine, international conflicts and relations, a forest, and the human body. Complex systems display properties that can be modelled by dynamical theory and need to be understood [17,18,19,20,21]. These include:

A: Emergence. A complex system has many parts which are distributed without a central controller. The parts are neither fixed in position as in a crystal, nor are they random as the molecules of gas. The organization of the system is derived from local interactions between the components as self-organization, a pattern which cannot be determined by a single component. This property of emergence is non-reducible. Focusing on the brain in isolation does not describe the movement that emerges when individuals and environments interact with each other. 

B: Nonlinearity. The complexity of interactions between component parts results in output states dependent on initial conditions, as synergies, cancellations, or steady state. Actions between agonist and antagonistic muscles may not move a limb. The component muscle fibers of a large muscle contract together to produce synergy as a force greater than the sum of each fiber. Nonlinearity also implies jumps, skips, and regressions observed in the learning process [22].

C: Autonomy and adaptation. Complex systems do not have centralized control for coordinating the entire system. The Internet, banking systems, and traffic control are examples of this property. The individual elements can adapt to the local environment according to its own set of instruction or rules. Eco-systems respond differently to local rainfall conditions. Similarly, anesthetists adapt to generate individual ways of holding a spinal needle during insertion during interaction with the needle. Practitioners hold intravenous cannulas in different ways depending on the shape, size, and contour of the cannula they use with respect to their own intrinsic dynamics. This adaptation tends to be through evolution rather than revolution as their holding patterns arise from the search for functional attractors.

D: Constraints. Complex systems are constrained by factors from outside and within at a particular time. Constraints define the boundaries or limits of the system states but can also be viewed as the factors that determine the possible states of the system. Some constraints are easier to understand as in gravity forcing planets to move in orbits, whereas others are complex as in explaining how physical size alters body position and force applied during external chest compressions. Constraints are physical and informational. A flock of birds flying in formation, and shoaling fish swarming in schools without colliding with each other, gain information from each other to generate their formations.

Intentional movements are a special subset of movements that exist in neurobiological systems. They are also limited by informational constraints over and above physical constraints [23]. The intention-perceiving-acting cycle represents the complex energy-information coupling between organisms and environments, residing only in the overlapping affordances set of nonstationary relational properties and effectivities (abilities) arising from goal constraints and action constraints [23]. This set satisfies “meaningful content” embodied in the complex dynamics of movement [24].

Thus, dynamical systems theory contributes to the understanding of movements as a complex system with properties of self-organization, and the emergence of movements. The system is limited by various interacting constraints to exhibit behaviors that form stable states in time. In goal-directed activity intentions form an informational constraint.

### 3.4. The ECD Explanation of Movement

In explaining goal-directed movement, ECD proposes that individuals initially generate an intention for the movement. An intention may be climbing six flights of stairs. The intention drives active perception from the environment of affordances. These may include the presence of a handrail for a person with knee problems. The climbing movement that arises from an interaction with the handrail and stairs includes weight distribution between the legs in moving. During climbing, there is a cyclical interaction between the perception of the number of steps left, step-height and whether they may need to stop and rest or keep going. If, on reaching the third floor, the pain in the knee becomes severe, the intention can change, and may opt to use a lift instead. Thus, intention drives perception-movement feedback loops to allow movement to emerge, limited by the constraints. 

### 3.5. ECD Perspective on Cognitive Structures or Processes

ECD provides alternate views on cognitive structures proposed in the cognitive control model including cognition, schemas, memory, and decision-making, in motor-skills without the requirement of representation and motor plans.

Cognition has traditionally been defined as the information processing that produces mental representations [25]. The ECD defines cognition as “the ongoing, active maintenance of a robust organism-environment system, achieved by closely co-ordinating perception and action” [26]. Cognition is embedded, embodied, enacted, and enhanced through the perception-movement interaction rather than a separate event for recall [27,28,29,30]. Cognition requires a body-brain-mind unit. The extracranial component for movement is the performer-environment unit [27]. ECD supports this concept. A learner understands the nature of a vein, the variation and regularity of its anatomical distribution, variations of depth, and relationships with arteries directly from performance. This cognitive knowledge leads to decision-making in a choice of location for venous cannulation. In performing chest compression, the knowledge of resistance to applied downward force can only be learnt through performance, even if it cannot be directly expressed in words.

ECD does not support the concept of movement memory. As each performance relies on the performer-environment interaction, memory does not play a direct role. ECD does not preclude other roles of memory. Performance is stored, if needed, as an episodic memory, then converted as long-term memory in semantic terms [31]. The goals and subgoals may be stored but are utilized in relation to perception from the environment. In performing an intubation, the goal of laryngoscopy is stored and used as an informational constraint on the movement. If the patient has a large amount of blood in the airway the semantic nature of laryngoscopy can be used to perceive laryngoscopy view for intubation.

Critical decision-making in medicine usually relates choosing between courses of action. It is well-recognized as a non-technical skill with components including identifying options, balancing risks and selecting options, and re-evaluating [32]. Traditional decision-making models proposed include pattern-matching, heuristics, expected value, and Bayesian probability [33]. The decisions are made from a conceptual model of knowledge consisting of rules, definitions, and principles [34]. Decision-making in a cognitive frame requires knowing what options to look for, is prone to bias, and mainly focuses on goals that can be set or adjusted. Decision-making within an action is not formally recognized, as following the cognitive processes outlined but is presented in a stored representation of the task in a set of if-then rules to be learnt. In ECD, decision-making behavior is explained at the performer-environment relationship as emerging from interactions within environmental constraints over time towards specific goals. Decisions are generated from perceived affordances prospectively controlling behavior by detecting informational constraints specific to goal-paths [2,35].

### 3.6. ECD Applied in Other Domains

The principles of ECD are a general explanation of how movements are generated. The functions of learning and acquisition from it can be generalized to many domains. These include sport, music, industrial, military, and human factors skills [36]. The most significant difference from cognitive approaches is the lack of a representation of a task, and the need for a motor-plan that can be recalled, and run-off as required. The environment provides enough information to allow direct perception and action without cognitive processing [8]. 

The concepts of ECD, and their manifestations via a Constraint-Led Approach (CLA) to coaching, are well established and extensively applied in sports. For example, compared with prescriptive instruction, participants learning to strike to the opposite field in baseball, performed better when trained with a constraints approach. In climbing, differences were noted in bodily movements between experts and novices related to constraints and differently perceived affordances [37]. In mathematics, having students first figure out how to enact the dynamic instantiation of a new concept by solving a bimanual motor-control problem, and only then introducing supplemental constraints on their perception-action loops, in the form of formal symbolic artifacts, such as a grid of lines, enables the students to bootstrap the new concept into conscious reflection [3]. In music, incorporating ECD principles allowed the development of music performance on the jazz-trumpet [38]. These concepts can be applied to healthcare procedural skills.

## 4. Application of ECD to Medical and Anaesthesia Tasks

Combining ecological psychology with dynamical systems theory leads to generation of the ecological-dynamic (ECD) framework (Figure 1).

Medical and anesthetic tasks, such as intubation, intravenous cannulation, the insertion of a spinal needle, laparoscopy, and gut endoscopy, are composed of identifiable sequential steps. Each step represents a sub-goal (intention). An active perception of affordances links in a continuous cycle with movements to achieve the goal. Movements emerge from an interaction of constraints within a complex system consisting of performer (learner), environment, and task.

Newell describes three types of physical constraints, task-, environmental-, and performer-related [40]. Informational constraints are another form of constraint and they have a significant role in intentional behavior [23]. A common informational constraint in medicine is patient safety with harm minimization. This restricts performing a spinal tap to below the level of spinal cord termination. Informational constraints regarding the task include its indications and contra-indications.

Performer constraints relate to physical attributes, such as size for external chest compressions or handling patients, dexterity, and co-ordination. Functional constraints relate to attention, motivation, or affective factors such as confidence or mood. The environment is composed of the patient, equipment, room, lighting, location, assistants, and other personnel including surgeons. Environmental constraints usually affect the movements made by the performer. The angle of approach of a cannula varies with the depth of the vein in patients, with shallower angles for superficial veins and steeper for deeper veins. The nature of a cannula changes the grip. Retractable intravenous cannulas are held differently to standard cannulas.

The process of performing a task begins with the generation of intentions or goal of a movement. This generates active perception from the environment. The individual picks-up information from the environment consisting of specific variables that allow the link to a movement. Unlike indirect perception, information is derived directly from the patterns of light reaching the retina [11]. In daily life, a person with an intention of going through a door gains specific information from the door frame, and the door handle, as to whether the door will need to be pushed or pulled open to achieve the goal. An anesthetist gains specific information about a patient’s spine to decide the approach of the epidural needle. A resuscitator gains specific information about a victim of a cardiac arrest, the location, and other team members, to generate how external chest compressions will be performed. Observations of learners performing compressions suggest rescuers of 50 kg weight use both upper body weight and force from the shoulders to generate the 5 cm compression depth compared with taller, heavier rescuers, who use upper body weight alone. The stance taken by the shorter rescuer’s knees is also broader to gain a stable base to apply extra force to achieve the target of 5 cm compression depth for on-floor resuscitation. 

Perception is active and is for affordances. In goal-directed behavior there is a continual search for the movement that can be made towards the intention. Specific variables related to the agent are utilized to appreciate affordances. Affordances relate to what is available to the performer from the environment in relation to their effectivities or capabilities. Perception is easier when moving than when standing still [13]. In ultrasound-guided cannulation, it is easier to understand the nature of the vessel and its size when moving the transducer than a single image in cross-section. Depending on the experience, coordination ability, the state of anxiety, the physician will choose the vein and cannula size (affordance as perceived by the performer). Cardiologists find it easier to understand valve function when the valve is moving, rather than from a single image.

A key point in ECD is the coupling of information to movement. Movement changes the environment, which changes the information provided to the performer. As a cannula is inserted under the skin, the physician appreciates resistance of the tissues, as an altered perception. The force applied by the hand on the cannula is varied in response. This perception-movement cycle controls the movement. The most relevant information for decision-making and the regulation of action in dynamic environments emerges during continuous performer-environment interactions [2]. The focus provided by intentions combined with experience allows the active perception of affordances to be achieved based on critical information variants in the environment, and the reciprocity shapes intentions and enhances decision-making, planning and organization of goal-directed activity [2]. During a video-laryngoscopic intubation of a patient with a difficult airway, the view obtained determines the need to use a bougie. In turn, the movement of the bougie generates a change in the laryngeal shape to alter the movements to achieve intubation.

## 5. Skill Development with ECD Principles

### 5.1. Basic Tenets of ECD to Understand Performance

Four basic tenets of ECD are proposed to understand performance to guide skill development. 

A: Movement variability has a functional role. In practice, most tasks performed by a physician are adjusted to the patient and environment. Intubation movements are adjusted based on neck mobility, mouth opening, location of intubation, and whether performed in an elective or emergency setting. ECD considers this adaptive behavior as a balance between movement pattern stability and flexibility relative to the performance context [10]. This dynamic, information-movement coupling relies on specifying variables actively perceived as affordances. Movement variability results from physical and informational constraints on the system. Skill development requires the learner to be exposed to multiple variations, allowing exploration, with expertise considered as the continuous adaptation to dynamic environments, displayed as appropriate balance between stable and variable movement patterns [41,42].

B: Skill development and transfer require representative learning and training design [2]. Complex adaptive systems need specific variables required for appropriate perception-movement coupling. Teaching emergency front-of-neck airway access on a manikin requires it (manikin) to represent patients with such difficulties rather than a ‘standard’ neck. The environment needs to signal an appropriate oxygen desaturation to represent the nature of the emergency in the learning phase. This tenet implies a need to analyze tasks by movements for specifying information variants, and constraints.

C: Attunement to affordances. Affordances are a key property of performer-environment interactions, that guide behaviors of complex systems [13]. Affordances are action opportunities related to the abilities of the performer (effectivities). They are not constant, but dynamic in a complex system. The performer needs perception of the environment rather than knowledge about the environment [10]. Skilled individuals become more attuned to available affordances, explaining why a skilled anesthetist is able to select and cannulate a vein which a novice does not choose. A large range of specifying variables from different sensory modalities need to be actively perceived in some cases to allow for the appropriate movement to emerge from a range of possible movements. Examples include the approach to performing an ultrasound sciatic nerve block, diagnosing cardiac function on a trans-thoracic ultrasound, and bronchoscopy.

D: Non-linear pedagogy. Neurobiological complex systems display non-linear outputs, with attractors and repellers. Learning becomes non-linear in these systems. This implies task simplification rather than decomposition, to allow the learner to explore the information-movement coupling and generate experience-based information constraints on movement [43]. There is non-proportionality between the amount of practice and skill development, emphasizing the need to focus on elements of developing information-movement coupling, and the manipulation of constraints [10].

### 5.2. Skill Acquisition in ECD

Learning in an ECD framework aims to establish the information-movement coupling resulting from interacting constraints on the performer-environment-task complex. Three key processes are required: education of intentions; attunement to environment; and the calibration of action to critical information in the environment. 

Learners need to be educated in intentions [2]. In ultrasound-guided peripheral vascular cannulation one step is choosing an appropriate vein. This intention guides the performer to harness information sources to support their choice. Information sources include ultrasound image, patient forearm position, the presence of scars and burns (visual information), touch, or feeling skin and tissue resistance.

The second key goal is attunement to environment, with a continuous co-regulation of perception and movement [44]. This requires education of attention [8]. Attunement can take place through instruction, scaffolding with a peer or more experienced person, and personal experience. 

The third key element is the calibration of action by tuning movement to critical information source to regulate behavior [39]. The trainee learns to focus on key aspects that guide the movements.

### 5.3. Implications for Training

Adopting an ECD framework changes the nature of training and requires the teacher to understand the movement analysis of a task, constraints that can be manipulated, embedding cognition through movement, and ensuring the key elements described above. This constraint-led approach to practical and procedural skill teaching is yet to be considered in healthcare.

The concepts proposed are founded on multiple seminal works. They include complex system behavior including synergies, degeneracy, self-organization, and multistable coordination dynamics by Kelso [45], perception-action coupling by Gibson [13], and ideas of motor control by Bernstein [46].

## 6. Conclusions

Current training in healthcare is built on educational models that focus on cognitive approaches to skills acquisition. Domains such as sport have demonstrated that participants learn and master skills faster using training developed through ECD. The similarities between the skills developed in sports and many of the clinical skills required in healthcare is clear. Therefore, healthcare needs to investigate adopting the ECD framework to understand task performance, skill acquisition, and its development. This wider application is supported by manuscripts addressing the functions of learning [36]. However, the change requires considerable adjustment from the current cognitive approaches of educators.

## Figures and Tables

**Figure 1 healthcare-11-00038-f001:**
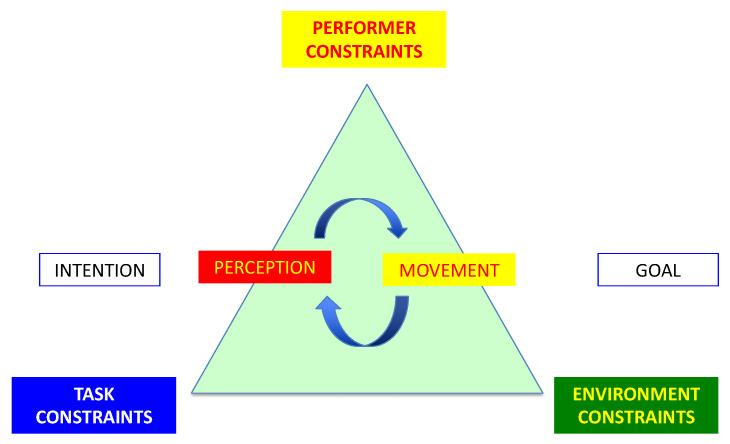
The ecological-dynamic model (Redrawn from the constraints-based approach to motor learning: Implications for a non-linear pedagogy in sport and physical education [39]).

## Data Availability

Not applicable.

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
