# Peer review of "The Ecological-Dynamics Framework for Medical Skills"

_healthcare, 2022, doi:10.3390/healthcare11010038_

Round 1
Reviewer 1 Report
healthcare-2117373
In this conceptual paper, the authors argue for the theoretical and practical utility of applying an ecological-dynamics view of medical skill performance, with implications for pedagogy. I find the thesis timely, compelling, and cogent. I offer only a few comments, one proposed improvement, and some sporadic language points.
The paper opens with a dense overview of ecological dynamics, which could be daunting for the uninitiated yet, given repeated readings and, possibly, reference to the cited works, could bring people on the right track. After all, this is a paradigm shift, and even though cybernetics has been with us for almost a century, it goes counter not only to mainstream cognitive science but also to folk epistemology, linguistic syntax, and, hence, tacit Cartesian sense of ontologies as “things out there.”
In discussing complex dynamic systems in flux, the authors should probably cite the seminal work of Scott Kelso, e.g.,
Kelso, J. A. S. (1995). Dynamic patterns: The self-organization of brain and behavior. MIT Press.
Or perhaps
Kelso, J. A. S. (2000). Principles of dynamic pattern formation and change for a science of human behavior. In L. Lars, R. Bergman, R. B. Cairns, L. G. Nilsson, & L. Nystedt (Eds.), Developmental science and the holistic approach (Proceedings of a conference at Wiks Castle and the Nobel Institute, Stockholm, Sweden) (pp. 63–83). Erlbaum.
I also thought that the absolutely foundational historical work of Nikolai Bernstein deserves at least a ‘shout out,’ e.g.., the Turvey/Latash 1996 book. His ideas, such as skill learning as the development of automatisms, or “repetition without repetition,” are absolutely thematic to ECD.
Latash, M., & Turvey, M. T. (Eds.), Dexterity and its Development. Lawrence Erlbaum Associates.
p. 5, may I suggest the following rephrasing:
“In mathematics, having students first figure out how to enact the dynamic instantiation of a new concept by solving a bimanual motor-control problem, and only then introducing supplemental constraints on their perception-action loops, in the form of formal symbolic artifacts, such as a grid of lines, enables the students to bootstrap the new concept into conscious reflection [3]”
Some language points
p. 1 “Over the last four decades movement psychologists have adopted concepts of ECD to explain…” When the acronym “ECD” is first introduced, we need the whole phrase.
p. 1 “This article proposes an ecological-dynamics model of movements is the framework” change “is” to “as” Again, in “Hypothesis,” it’s strange to use “is” – prefer “as”
p. 4, removed second “if” in “If on reaching the third floor the pain in the knee becomes severe…” Pay attention also to the second clause in this sentence, which is quite non-normative (“and they opt to use a lift instead”)
p. 4 (immediately after) “intention drives perception-movement feedback loop[s]…”
p. 5, “CLA” first appears. Make sure we know what this means.
p. 6 This sentence comes across as a logically not as compelling as it might be: “This perception-movement cycle controls the movement.”
p. 7 (bottom): You enumerate three goals. See if you can fix the paragraph formatting to help the reader see the structure, e.g. start new paragraph for each of the goals, or make these into bullets.
Author Response
Dear Reviewers,
Thank you for reviewing the document on Ecological-Dynamics titled “The Ecological-Dynamics Framework for Medical Skills” and considering it for publication.
Reviewer #1:
The changes suggested had been made and marked by the track changes method. These include:
- 5, may I suggest the following rephrasing:
“In mathematics, having students first figure out how to enact the dynamic instantiation of a new concept by solving a bimanual motor-control problem, and only then introducing supplemental constraints on their perception-action loops, in the form of formal symbolic artifacts, such as a grid of lines, enables the students to bootstrap the new concept into conscious reflection [3]”
Some language points
- “Over the last four decades movement psychologists have adopted concepts of ECD to explain…” When the acronym “ECD” is first introduced, we need the whole phrase. – Done p.1: Ecological-Dynamics (ECD)
- “This article proposes an ecological-dynamics model of movements is the framework” change “is” to “as” Again, in “Hypothesis,” it’s strange to use “is” – prefer “as”
- removed second “if” in “If on reaching the third floor the pain in the knee becomes severe…” Pay attention also to the second clause in this sentence, which is quite non-normative (“and they opt to use a lift instead”)
- (immediately after) “intention drives perception-movement feedback loop[s]…”
- “CLA” first appears. Make sure we know what this means. P.5: Constraint-Led Approach (CLA)
- This sentence comes across as a logically not as compelling as it might be: “This perception-movement cycle controls the movement.”
- (bottom): You enumerate three goals. See if you can fix the paragraph formatting to help the reader see the structure, e.g., start new paragraph for each of the goals, or make these into bullets. Done – formatted in 3 paragraphs
There are many seminal works regarding ecological dynamics as the material is well established in the last two decades of the twentieth century and adopted by sports and physical education. There are many authors who have provided ground-breaking innovations making it difficult to name them all without generating a historical review. We believe the theory should also apply to medical skills which is why this first article is produced. In the modification of the article, we have included Kelso JAS, Gibson JJ and referred to Bernstein who challenged the cognitive models with computer analogies.
Two extra references are added as suggested by reviewer #1:
- Kelso J. A. (2012). Multistability and metastability: understanding dynamic coordination in the brain. Philosophical transactions of the Royal Society of London. Series B, Biological sciences, 367(1591), 906–918. https://doi.org/10.1098/rstb.2011.0351
- Turvey, M.T. (1978). Issues in the theory of action : Degree of freedom, coordinative structures and coalitions. Attention and Performance, 557-595.
Yours sincerely,
Reviewer 2 Report
I find the article interesting and the authors make a compelling plea for the adoption of the Ecological-Dynamics Framework to develop medical skills. However, I suggest the authors either place the article as an opinion piece, or add a discussion concerning alternative paradigms. imply stating that the current cognitive-based approach to training has limitations seems not enough. As it stands now, throughout the article the authors discuss that ECD is appropriate and useful. As a conclusion, authors state that the change in adopting ECD "requires considerable adjustment from current approaches for educators". This statement needs better unfolding, rather than statement. It would have been of help to see further development of the topic, e.g. the testing of the efficacy of this paradigm in an experimental cohort of students or an investigation concerning the readiness/obstacles to introduce this approach in the medical training.
Author Response
Dear Reviewers,
Thank you for reviewing the document on Ecological-Dynamics titled “The Ecological-Dynamics Framework for Medical Skills” and considering it for publication.
Reviewer #2:
We have altered the statement regarding current approaches to include the term “cognitive” although a formal comparison and reasons for change would need another article. The testing of this paradigm is the subject of another paper being produced as the experimental component of the PhD thesis (Kersi Taraporewalla) comparing the efficiency and efficacy of a CLA approach to current teaching.
We hope these alterations satisfy you to provide support for publication of the paper.
Thank you for your efforts and support.